# Development and External Validation of a PET Radiomic Model for Prognostication of Head and Neck Cancer

**DOI:** 10.3390/cancers15102681

**Published:** 2023-05-09

**Authors:** Wyanne A. Noortman, Nicolas Aide, Dennis Vriens, Lisa S. Arkes, Cornelis H. Slump, Ronald Boellaard, Jelle J. Goeman, Christophe M. Deroose, Jean-Pascal Machiels, Lisa F. Licitra, Renaud Lhommel, Alessandra Alessi, Erwin Woff, Karolien Goffin, Christophe Le Tourneau, Jocelyn Gal, Stéphane Temam, Jean-Pierre Delord, Floris H. P. van Velden, Lioe-Fee de Geus-Oei

**Affiliations:** 1Section of Nuclear Medicine, Department of Radiology, Leiden University Medical Center, 2333 ZA Leiden, The Netherlands; 2TechMed Centre, University of Twente, 7522 NB Enschede, The Netherlands; 3Nuclear Medicine Department, Centre Hospitalier Universitaire de Caen, 14000 Caen, France; 4Technical Medicine, Delft University of Technology, 2628 CD Delft, The Netherlands; 5Amsterdam University Medical Center, 1081 HV Amsterdam, The Netherlands; 6Department of Biomedical Data Sciences, Leiden University Medical Center, 2300 RC Leiden, The Netherlands; 7Nuclear Medicine and Molecular Imaging, Department of Imaging & Pathology, University Hospitals Leuven, KU Leuven, 3000 Leuven, Belgium; 8Department of Medical Oncology, Institut Roi Albert II, Cliniques Universitaires Saint-Luc, 1200 Brussels, Belgium; 9Institute for Experimental and Clinical Research (IREC, pôle MIRO), Université Catholique de Louvain (UCLouvain), 1200 Brussels, Belgium; 10Department of Head and Neck Medical Oncology, Fondazione IRCCS Istituto Nazionale dei Tumori, University of Milan, 20133 Milan, Italy; 11Division of Nuclear Medicine, Institut de Recherche Clinique, Cliniques Universitaires Saint Luc, 1200 Brussels, Belgium; 12Department of Nuclear Medicine-PET Unit, Fondazione IRCCS Istituto Nazionale dei Tumori, 20133 Milan, Italy; 13Nuclear Medicine Department, Institut Jules Bordet, Université Libre de Bruxelles (ULB), Hôpital Universitaire de Bruxelles (H.U.B.), 1070 Bruxelles, Belgium; 14Department of Drug Development and Innovation, Institut Curie, Paris-Saclay University, 75005 Paris, France; 15Epidemiology and Biostatistics Department, Centre Antoine Lacassagne, University Côte d’Azur, 06100 Nice, France; 16Department of Head and Neck Surgery Gustave Roussy, 94805 Villejuif, France; 17Department of Medical Oncology, IUCT-Oncopole, 31100 Toulouse, France; 18Department of Radiation Science & Technology, Delft University of Technology, 2628 CD Delft, The Netherlands

**Keywords:** [^18^F]FDG PET/CT, head and neck squamous cell carcinoma, radiomics, machine learning, overall survival, afatinib

## Abstract

**Simple Summary:**

A lack of external validation is still one of the major limitations of radiomics, hampering its clinical translation. The aim of this study was to build and externally validate an [^18^F]FDG PET radiomic model to predict overall survival in patients with head and neck squamous cell carcinoma treated with preoperative afatinib. Radiomic analysis of two cohorts of 20 and 34 patients was performed, where each cohort served once as a training and once as an external validation set. The radiomic model was compared to a clinical model and to a model that combined clinical and radiomic features. The radiomic model surpassed the clinical model in terms of predictive performance, but the combination of the radiomic and clinical model performed best. The [^18^F]FDG-PET radiomic signature based on the evaluation scan seems promising for the prediction of overall survival in HNSSC treated with preoperative afatinib.

**Abstract:**

Aim: To build and externally validate an [^18^F]FDG PET radiomic model to predict overall survival in patients with head and neck squamous cell carcinoma (HNSCC). Methods: Two multicentre datasets of patients with operable HNSCC treated with preoperative afatinib who underwent a baseline and evaluation [^18^F]FDG PET/CT scan were included (EORTC: n = 20, Unicancer: n = 34). Tumours were delineated, and radiomic features were extracted. Each cohort served once as a training and once as an external validation set for the prediction of overall survival. Supervised feature selection was performed using variable hunting with variable importance, selecting the top two features. A Cox proportional hazards regression model using selected radiomic features and clinical characteristics was fitted on the training dataset and validated in the external validation set. Model performances are expressed by the concordance index (C-index). Results: In both models, the radiomic model surpassed the clinical model with validation C-indices of 0.69 and 0.79 vs. 0.60 and 0.67, respectively. The model that combined the radiomic features and clinical variables performed best, with validation C-indices of 0.71 and 0.82. Conclusion: Although assessed in two small but independent cohorts, an [^18^F]FDG-PET radiomic signature based on the evaluation scan seems promising for the prediction of overall survival for HNSSC treated with preoperative afatinib. The robustness and clinical applicability of this radiomic signature should be assessed in a larger cohort.

## 1. Introduction

Head and neck squamous cell carcinoma (HNSCC) is a common type of cancer worldwide, with approximately 900,000 cases and over 400,000 deaths in 2020 [1]. The oral cavity, oropharynx, larynx, and hypopharynx are the most frequently affected sites. Treatment options depend on disease stage and location and include surgery, radiation therapy, chemotherapy, and targeted therapy, either alone or in combination [2]. Despite advances in treatment, the prognosis for patients with advanced HNSCC remains poor, especially in advanced disease stages [3]. Recently, the tyrosine kinase inhibitor afatinib, a form of targeted therapy, has demonstrated efficacy in various types of cancer, including HNSCC [4]. In addition, its role in previously untreated HNSCC patients has been studied, where 2-[^18^F]fluoro-2-deoxy-D-glucose positron emission tomography ([^18^F]FDG PET) was used to quantify the metabolic response to afatinib therapy [5]. However, [^18^F]FDG PET images might contain more information than can be assessed visually or using traditional quantitative metrics. Quantitative image analysis using radiomics may provide more insights into disease biology. Radiomics is a form of medical image processing that aims to find stable and clinically relevant image-derived biomarkers for lesion characterisation, prognostic stratification, and response prediction, thereby contributing to precision medicine [6,7]. Radiomics consists of the conversion of (parts of) medical images into a high-dimensional set of quantitative features and the subsequent mining of this dataset for information useful for the quantification or monitoring of tumour or disease characteristics in clinical practice.

The field of radiomics has gained increasing attention in recent years, bringing about many proof-of-concept studies that demonstrate the feasibility and potential utility of radiomics in a wide range of applications. However, a lack of external validation is still one of the major limitations of radiomics, hampering its clinical translation. While the field of radiomics has grown and the number of papers has gradually increased over the last decade [8], widespread clinical implementation has not yet been established due to a lack of standardisation in the different steps of the radiomic pipeline and validation of the radiomic models. Individual studies show promising results, but they sometimes present spurious model interpretations caused by “cherry picking”, i.e., carrying forward only the best results obtained with a single approach while omitting contradictory results obtained with a slightly different approach [9]. Therefore, model performance should be validated using a new, independent dataset to assess whether the model is predictive for an entire target population or for a specific subset of patients only, e.g., patients with a specific demographic profile or patients who underwent a scan with a specific scanner or site-specific scan protocol. Validation using data that have neither contributed to the model design nor been kept aside from the original data set is highly recommended, preferably using external data from different institutes [10]. External validation is of utmost importance for the clinical implementation of radiomic models.

The aim of this study was to build and externally validate an [^18^F]FDG PET radiomic model to predict overall survival in patients with HNSCC treated with preoperative afatinib followed by surgery.

## 2. Materials and Methods

### 2.1. Patient Population, Data Acquisition, and Image Reconstruction

[^18^F]FDG PET scans of two multicentric prospective cohorts (European Organisation for Research and Treatment of Cancer (EORTC) study 90,111; Unicancer Predictor) of patients with previously untreated histologically or cytologically confirmed squamous cell carcinoma of the oral cavity, oropharynx, larynx or hypopharynx were retrospectively analysed [5,11]. Patients in the treatment arm, who received preoperative afatinib (40 mg/day) for 14 to 28 days (14 days in EORTC cohort, 28 days in Unicancer cohort) and underwent [^18^F]FDG PET imaging on day 0 (baseline) and day 15 (evaluation) followed by surgery, were included.

In the EORTC study (ClinicalTrials.gov Identifier: NCT01538381) [5], immunohistochemical staining was performed on 4 µm paraffin-embedded tumour sections, and p16 positivity was determined using a histology score of ≥210 according to Ang et al. (strong nuclear and/or cytoplasmic staining in ≥70% of tumour cells, Clone E6H4, CINtec^®^, Roche, Tucson, AZ, USA) [12]. PET/CT scanners were required to have EANM Research Ltd. (Vienna, Austria) (EARL) accreditation, and sites had to submit a dummy run scan following the study imaging requirements. PET acquisition was started 66 ± 6 min after the injection of 4.0 ± 0.7 MBq [^18^F]FDG per kilogram of body weight administered as an intravenous bolus. Serum glucose levels were below 7.0 mM/L. The reconstructed voxel sizes ranged from 2.73 × 2.73 × 3.27 mm^3^ to 4.00 × 4.00 × 4.00 mm^3^.

In the Unicancer study (ClinicalTrials.gov Identifier: NCT01415674) [11], HPV16 was detected by single PCR with specific HPV16 primers. Other HPV subtypes were determined using multiplexed PCR on DNA extracted from frozen or formalin-fixed paraffin-embedded tumour tissue samples. Baseline and evaluation PET scans were acquired on the same scanner using similar acquisitions and reconstruction settings. PET acquisition was started 68 ± 11 min after the injection of 3.75 ± 0.88 MBq [^18^F]FDG per kilogram of body weight administered as an intravenous bolus. Serum glucose levels were below 9.0 mM/L. The reconstructed voxel sizes ranged from 1.17 × 1.17 × 3.27 mm^3^ to 5.47 × 5.47 × 5.00 mm^3^.

All patients who entered the study signed an informed consent form. Both studies were conducted in accordance with the International Conference on Harmonization Good Clinical Practice standards and the Declaration of Helsinki. The EORTC trial was approved by Belgian and Italian ethics committees on 7 June 2012 and 17 July 2012, respectively, and by the Belgian and Italian health authorities on 7 May 2012 and 10 September 2012, respectively. The Predictor trial (Unicancer) was approved by a French ethics committee (Comité de Protection des Personnes Ile de France 1) on 12 July 2011 (reference 2011-mai-12618) and by the French health authorities ANSM on 21 June 2011 (reference A110602-11).

### 2.2. Quantitative Image Analysis

#### 2.2.1. VOI Delineation

Volumes of interest (VOI) delineation was performed using the Accurate tool (AmsterdamUMC, Amsterdam, The Netherlands) [13] on both the baseline and evaluation [^18^F]FDG PET scans using an isocontour that applies an adaptive threshold of 50% of the peak standardised uptake value (SUV_peak_), obtained using a sphere 12 mm in diameter [14], corrected for local background [15]. Boxing was applied to exclude surrounding [^18^F]FDG-avid tissues such as Waldeyer’s tonsillar ring. VOIs were delineated by W.A.N. (PhD candidate in PET radiomics with 5 years of experience) and supervised by D.V. (nuclear medicine physician with 14 years of experience).

#### 2.2.2. Radiomic Feature Extraction

Radiomic feature extraction was performed in PyRadiomics version 3.0 in Python version 3.7 (Python Software Foundation, Wilmington, DE, USA) [16]. For baseline as well as evaluation scans, 105 radiomic features were extracted: 18 first order features, 14 shape features, and 73 texture features (22 grey level co-occurrence matrix (GLCM), 16 grey level run length matrix, 16 grey level size zone matrix, 14 grey level dependence matrix, and 5 neighbouring grey tone difference matrix). In addition, total lesion glycolysis—the product of the mean SUV and the metabolic tumour volume—was calculated. A fixed bin size of 0.5 g/mL was applied, and images were interpolated to isotropic voxels of 4.00 × 4.00 × 4.00 mm^3^ using B-spline interpolation, with grids aligned by the input origin. In addition, delta radiomics expressing the percentual differences between baseline and evaluation features were calculated using Equation (1) [17]:(1)$$\Delta \mathrm{feature}=\frac{{\mathrm{feature}}_{\mathrm{evaluation}}-{\mathrm{feature}}_{\mathrm{baseline}}}{{\mathrm{feature}}_{\mathrm{baseline}}}\times 100\%$$

### 2.3. Statistical Analysis

Statistical analysis was performed in R version 3.6.0 (R Foundation for Statistical Computing, Vienna, Austria) using the packages FMRadio and randomForestSRC [18,19]. Differences in clinical characteristics between cohorts were compared using Pearson’s chi-squared test for categorical variables and the Mann–Whitney U test or the independent sample t-test after testing for (log-)normality for continuous variables. Differences in survival curves were compared using log-rank statistics. As the cut-off for considering statistical significance, a type I error probability (p) of 0.05 was defined for each test.

Each cohort served once as a training set and once as an external validation set for the prediction of overall survival. Overall survival was measured from the date of inclusion to the date of death by any cause or was censored to the date of last contact. Features were standardised (centred around 0, standard deviation of 1) to prevent features with the largest scale from dominating the analysis. Redundancy filtering of the Pearson correlation matrix (r = 0.9) was performed. Subsequently, supervised feature selection was performed using variable hunting with variable importance [20], which was repeated 1000 times, selecting the top two features (i.e., one feature per ten subjects) ranked in terms of occurrence. A Cox proportional hazards regression model using the selected radiomic features was fitted on the training dataset and validated in the external validation set. In addition, a clinical model that univariately selected the two best-performing clinical characteristics, and a combined model that combined the two radiomic features and the two clinical variables, were fitted. Model performances are expressed by the concordance index (C-index). The C-index is a generalisation of the area under the ROC curve that can take into account censored data. The proportional hazard assumption was graphically assessed by plotting the Schoenfeld residuals against time.

The findings were validated in a sham experiment [21]. The outcome labels were randomly shuffled for 100 iterations, and mean C-indices were calculated. The randomisation of the outcome labels preserves the distributions and multicollinearity of the radiomic features and the prevalence of the outcome, but it uncouples their potential relation.

## 3. Results

Twenty and thirty-four patients were included in the EORTC and Unicancer cohort, respectively (Figure 1). Three patients were excluded because no evaluation scan was available. Clinical characteristics and maximum SUV (SUV_max_) were not significantly different between cohorts (Table 1). The metabolic tumour volumes at baseline PET as well as at evaluation PET were significantly larger in the Unicancer cohort compared to the EORTC cohort (Table 1). Plotting the Schoenfeld residuals against time showed that all the covariates in the Cox proportional hazards model met the proportional hazard assumption (Appendix A).

When the EORTC dataset served as a training set and the Unicancer dataset as an external validation set, the clinical model alone yielded a C-index of 0.60 in the validation set for the prediction of overall survival (Table 2). The selected clinical characteristics were the age of the patient and N-stage. The radiomic model performed slightly better, with a C-index of 0.69 in the validation set. Two shape features from the evaluation PET were selected: the sphericity and the maximum 2D diameter of the slice. The combined model, including both the selected clinical features and the selected radiomic features, performed similarly, with a C-index of 0.71 in the validation set.

In contrast, when the Unicancer dataset served as a training set and the EORTC dataset as an external validation set, the clinical model alone yielded a validation C-index of 0.67 (Table 3). The selected clinical characteristics were again the age of the patient and N-stage. The radiomic model performed better, with a validation C-index of 0.79, selecting two shape features from the evaluation PET: the sphericity and the maximum 2D diameter of the column. The combined model performed slightly better, with a validation C-index of 0.82.

In the sham experiment, no model yielded a C-index different from 0.5 (range: 0.47–0.51) in the validation set.

## 4. Discussion

In this study, we built and externally validated a radiomic model with [^18^F]FDG PET features to predict overall survival in patients with HNSCC treated with preoperative afatinib prior to surgery. Even though we analysed two small cohorts, model development and validation seem feasible. The radiomic models surpassed the clinical models in terms of predictive performance, but the combination of the radiomic and the clinical model showed the best performance.

HNSCC is a challenging disease to treat, particularly when it is diagnosed at an advanced stage, and early diagnosis and intervention are crucial for improving patient outcomes [3,22]. Radiomics has the potential to play an important role in the management of HNSCC by identifying patients who may benefit from a specific therapy, evaluating the risk of recurrence, and predicting survival. Several studies assessed [^18^F]FDG PET radiomics in HNSCC for numerous applications [23,24,25,26]. In addition, many studies have investigated radiomics extracted using different modalities to assess the effectiveness of various tyrosine kinase inhibitors in different cancer types [27,28,29,30,31,32]. However, to the best of our knowledge, no study specifically assessed [^18^F]FDG PET radiomics for the prognostication of HNSCC patients.

Interpretation of the radiomic features may provide insight into the semantics or tumour phenotype as captured by the PET scan. In both models, the shape feature sphericity in the evaluation PET was selected. This indicates that sphericity is a robust feature, since radiomic feature selection generally does not return identical features between methods or sometimes even between folds due to the large number of (multicollinear) features. Sphericity is a measure of the roundness of the VOI relative to a sphere. The value ranges from 0 to 1, where a value of 1 indicates a perfect sphere. A lower sphericity, i.e., a larger surface compared to the volume, is associated with higher mortality. It is hypothesised that the sphericity of the tumour impacts its resectability: a perfectly spheric tumour might be easier to resect than an aspheric tumour with a large surface area compared to its volume. Several previous studies already described the prognostic value of primary tumour *asphericity* in (oral cavity) HNSCC determined from [^18^F]FDG PET [33,34] and CT [35], including cut-offs for the prediction of overall and progression-free survival. Please note that the definition of sphericity and asphericity may vary between software packages, and cut-offs might not be applicable for each software implementation. Preferably, feature definitions of the Image Biomarker Standardisation Initiative would be used [36].

In addition, both models selected a 2D diameter feature from the evaluation PET, which is defined as the largest pairwise Euclidian distance in a specific direction. The 2D diameter slice represents the diameter in the axial plane, and the 2D diameter column represents the coronal plane. This indicates that a larger tumour diameter, specifically in the transversal plane, is associated with higher mortality. Tumour size is considered an important prognostic factor in HNSCC [22,37]. Our findings are similar to those of Apostolova at al., as they present an incremental prognostic value of metabolic tumour volume (tumour dimension in 3D) in addition to asphericity [33].

It should be noted that both radiomic models only selected shape features. This might be attributed to the fact that the recommended minimal number of voxels for the texture analysis of 64 voxels per VOI was not reached in around one third of the VOIs [9]. For cubic regions, this corresponds to 4 × 4 × 4 mm^3^ voxels, i.e., ~4 cm^3^. Consequently, in terms of tumour size, HNSCC are not the most suitable lesions for PET radiomic analysis. Another possible explanation for the selection of two shape features might be the heterogeneity of the datasets. We investigated two multicentre studies that included scans from four and eleven different centres, respectively. While the EORTC centres were EARL-accredited, no such standardisation and/or harmonisation was performed in the Unicancer cohort. Radiomic features are highly impacted by image acquisition and reconstruction settings [38]. A multicentre phantom study by Pfaehler et al. in three institutions using six PET/CT systems and using clinical as well as EARL1 and EARL2 reconstructions demonstrated that the percentage of intensity and texture features that yielded a moderate or better (>0.6) intraclass correlation was low when compared to shape features (~30% and ~20% vs. ~90%, respectively) [39]. Moreover, they showed that EARL-compliant (preferably EARL2) reconstructions are beneficial and lead to a larger number of reliable, repeatable, and reproducible features.

Post-reconstruction harmonisation strategies such as ComBat harmonisation could have been employed to reduce heterogeneity and deal with the batch effect (i.e., the centre effect or imaging protocol effect) [40]. ComBat harmonisation is a method initially described for genomic data analysis that normalises radiomic features in order to remove the centre effect while retaining pathophysiologic information to in turn facilitate multicentre studies and exchange radiomic models between institutes. Unfortunately, with fifteen different imaging sites, the number of patients per centre was lower than the recommended twenty to thirty patients required for ComBat harmonisation [41]. ComBat harmonisation of the non-EARL-compliant imaging protocols onto the EARL-compliant imaging protocols could also be considered. However, ComBat harmonisation in prospective settings is challenging since the transformation can only be deduced from previously acquired data for the same patient population and imaging protocol [41], while in clinical practice, scanners and thereby imaging protocols are regularly replaced. In that perspective, selecting robust features irrespective of imaging protocol might be of greater interest. Since both models returned similar features, the radiomic model is considered relatively robust.

In addition, both models only selected features from the evaluation scan, disregarding the baseline and delta features. Evaluation features might be selected instead of baseline features, since evaluation features take into account the effect of the preoperative afatinib. Moreover, the delta radiomics could have been influenced by the heterogenous scanning protocols. Although little is known about the standardisation and technical challenges of delta radiomics [42], minor differences in patient preparation, acquisition, and reconstruction settings between the baseline and evaluation PET may have introduced errors.

It is somewhat surprising that the C-indices of the EORTC validation set are higher than those of the Unicancer training set, whereas the C-index of the training set is generally higher than the C-index of the validation set. A possible explanation might be that the EORTC validation set has less complexity, which might be due to the heterogeneity in the Unicancer dataset. This explanation might be confirmed by the fact that similar features were selected in both the EORTC and Unicancer model and by the fact that the model trained by the EORTC dataset shows a higher C-index than the model trained by the Unicancer dataset.

Our study has several strengths and limitations. The main strength of our work is the external validation, which most radiomic models lack. We also showed that model development and validation is possible in small datasets in the case of a strong biological signal. A limitation of this study is that we chose not to assess low-dose CT radiomic features since we already included over 300 baseline, evaluation, and delta features in two small datasets of 20 and 34 patients, respectively. Another limitation is that clinical parameters were not always assessed in a similar way across both studies, i.e., p16 positivity was considered a surrogate for HPV status. Although they are highly correlated, subgroups of p16+/HPV− and p16−/HPV+ can be identified with different prognoses [43]. In addition, treatment response to afatinib might have been an interesting outcome measure, but this was assessed as metabolic response in one study and according to RECIST criteria in the other. Similarly, progression-free survival was only monitored in one study.

## 5. Conclusions

Although assessed in two small but independent cohorts, an [^18^F]FDG-PET radiomic signature based on the evaluation scan seems promising for the prediction of overall survival from HNSSC treated with preoperative afatinib. The radiomic model surpassed the clinical model in terms of predictive performance, but the combination of the radiomic and clinical model performed best. The robustness and clinical applicability of this radiomic signature should be assessed in a larger cohort.

## Figures and Tables

**Figure 1 cancers-15-02681-f001:**
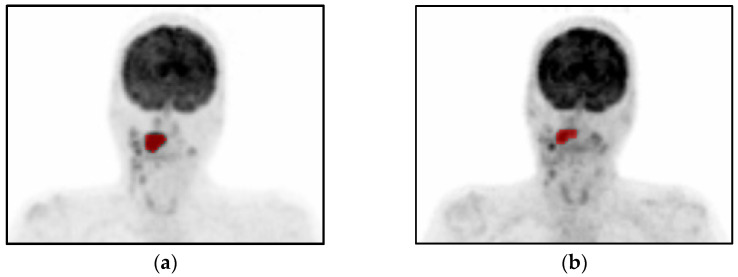
Example of [^18^F]FDG PET imaging in a patient with a HNSCC in the tongue (T2N2M0) with (**a**) the baseline scan and (**b**) the evaluation scan after preoperative afatinib (40 mg/day) for 14 days. Volumes of interests are indicated in red.

**Table 1 cancers-15-02681-t001:** Clinical characteristics and traditional quantitative imaging parameters of included patients.

Characteristic	EORTC (N = 20)	Unicancer (N = 34)	*p*-Value
Age (years), median (range)	58 (35–76)	57 (44–74)	0.911
Sex (M/F)	14/6	26/8	0.600
p16/HPV status * (positive/negative)	2/18	5/29	0.619
T-stage - T1 - T2 - T3 - T4			0.097
	2	1	
	12	11	
	1	4	
	5	18	
N-stage - N0 - N+			0.932
	8	14	
	12	20	
48-month survival (percentage ± standard deviation)	57.3 ± 13.2%	73.5 ± 7.6%	0.434 ^‡^
SUV_max_ baseline (g/mL), median (range)	10.6 (5.7–17.4)	12.2 (4.8–25.2)	0.257
SUV_max_ evaluation (g/mL), median (range)	7.1 (4.2–15.4)	8.5 (3.6–24.4)	0.252
MTV baseline (cm^3^), median (range)	4.9 (1.9–19.3)	7.9 (1.6–42.2)	0.032 ^†^
MTV evaluation (cm^3^), median (range)	4.2 (1.6–9.5)	5.4 (2.0–39.2)	0.041 ^†^

* p16 positivity (EORTC cohort) was considered a surrogate of HPV status (Unicancer cohort). ^†^ Significant. ^‡^ Log-rank. SUV_max_: maximum standardised uptake value, MTV: metabolic tumour volume.

**Table 2 cancers-15-02681-t002:** C-indices of the Cox proportional hazards model of the clinical model, the radiomic model, and the combined model. The EORTC dataset served as a training set, and the Unicancer dataset served as an external validation set.

C-Indices ± Standard Error	Training (EORTC)	Validation (Unicancer)
Clinical model: - Age - N-stage	0.70 ± 0.13	0.60 ± 0.10
Radiomic model: - Sphericity (evaluation PET) - Maximum 2D diameter slice (evaluation PET)	0.75 ± 0.11	0.69 ± 0.09
Combined model: All variables specified above	0.79 ± 0.09	0.71 ± 0.08

**Table 3 cancers-15-02681-t003:** C-indices of the Cox proportional hazards model of the clinical model, the radiomic model, and the combined model. The Unicancer dataset served as a training set, and the EORTC dataset served as an external validation set.

C-Indices ± Standard Error	Training (Unicancer)	Validation (EORTC)
Clinical model: - Age - N-stage	0.61 ± 0.10	0.67 ± 0.10
Radiomic model: - Maximum 2D diameter column (evaluation PET) - Sphericity (evaluation PET)	0.71 ± 0.08	0.79 ± 0.09
Combined model: All variables specified above	0.73 ± 0.08	0.82 ± 0.07

## Data Availability

Any requests for raw and analysed data will be reviewed by a data release committee to verify whether the request is subject to any intellectual property or confidentiality obligations. Patient-related data and images in the paper were generated as part of a clinical trial and are subject to patient confidentiality. Any data and materials (for example, clinical data or imaging data) that can be shared will need approval from the EORTC and Unicancer. Any data shared will be de-identified. A signed data access agreement with the collaborator is required before accessing shared data. Requests should be made to the corresponding author (w.a.noortman@lumc.nl); response time will be within approximately 10 to 20 business days.

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
