# Peer review of "Development and External Validation of a PET Radiomic Model for Prognostication of Head and Neck Cancer"

_cancers, 2023, doi:10.3390/cancers15102681_

Round 1

Reviewer 1 Report

The manuscript is a study about the building and externally validation of an [ 18F]FDG PET radiomic model to predict overall survival in patients with head and neck squamous cell carcinoma. The topic is interesting but some improvements should be made before considering it for publication in Cancers.

First of all, but it is just a comment, as you declared, the patient cohorts are very small and you already know how important is a robust number of data to strenghten the results of radiomics analysis.

In the Introduction section, you should briefly describe what radiomics is, and what head and neck squamous cell carcinoma is, its prognosis and its main critical issues. You may already know that H&N is quite generical, and several clinical scenario are present in this category of cancer.

Furthermore, did you think of the possible flare effect or something similar after afatinib therapy?

Finally, again, in the discussion you should emphasize the clinical implications, in a "practical" way of adding the results of a radiomics analysis to the management of these patients.

Author Response

Reviewer 1:

The manuscript is a study about the building and externally validation of an [ 18F]FDG PET radiomic model to predict overall survival in patients with head and neck squamous cell carcinoma. The topic is interesting but some improvements should be made before considering it for publication in Cancers.

Thank you for your comments regarding our article. We appreciate your time and effort reviewing our article and your suggestions. Thanks to your comments, our manuscript has greatly improved.

First of all, but it is just a comment, as you declared, the patient cohorts are very small and you already know how important is a robust number of data to strenghten the results of radiomics analysis.

We completely agree that the patient cohorts are rather small compared to the minimum number of patients that is considered suitable for radiomic analysis of about 80 patients per cohort. However, we wanted to show that, in case of a well reflected biological signal and careful selection of methods, radiomic analysis is also feasible in smaller datasets. However, this work should be considered as a proof-of-concept and more research is warranted to validate our findings. We mentioned in our conclusion that more research in larger cohorts is warranted to validate our findings.

In the Introduction section, you should briefly describe what radiomics is, and what head and neck squamous cell carcinoma is, its prognosis and its main critical issues. You may already know that H&N is quite generical, and several clinical scenario are present in this category of cancer.

Thank you for your feedback regarding our introduction section. We have added an introduction on HNSCC and radiomics in line 68-88.

“Head and neck squamous cell carcinoma (HNSCC) is a common type of cancer worldwide with approximately 900.000 cases and over 400.000 deaths in 2020 [1]. The oral cavity, oropharynx, larynx, and hypopharynx are the most frequently affected sites. Treatment options depend on disease stage and location and include surgery, radiation therapy, chemotherapy, and targeted therapy, either alone or in combination [2]. De-spite advances in treatment, the prognosis for patients with advanced HNSCC remains poor, especially in advanced disease stages [3]. Recently, the tyrosine kinase inhibitor afatinib, a form of targeted therapy, has demonstrated efficacy in various types of cancer, including HNSCC [4]. Also, its role in previously untreated HNSCC patients has been studied, where 2-[18F]fluoro-2-deoxy-D-glucose positron emission tomography ([18F]FDG PET) was used to quantify the metabolic response to afatinib therapy [5]. However, [18F]FDG PET images might contain more information than can be assessed visually or using traditional quantitative metrics. Quantitative image analysis using radiomics may provide more insights into disease biology. Radiomics is a form of medical image processing that aims to find stable and clinically relevant image-derived biomarkers for lesion characterisation, prognostic stratification, and response prediction, thereby contributing to precision medicine [6,7]. Radiomics consists of the con-version of (parts of) medical images into a high-dimensional set of quantitative features and the subsequent mining of this dataset for information useful for the quantification or monitoring of tumour or disease characteristics in clinical practice.

The field of radiomics has gained increasing attention in recent years, bringing about many proof-of-concept studies, which demonstrate the feasibility and potential utility of radiomics in a wide range of applications. The field of radiomics has gained increasing attention in recent years, bringing about many proof-of-concept studies, which demonstrate the feasibility and potential utility of radiomics in a wide range of applications. […]  “

Furthermore, did you think of the possible flare effect or something similar after afatinib therapy?

Disease flare is occasionally described when discontinuing (EGFR-) TKI treatment, in contrast to immunotherapy where hyper/pseudo progression is seen early after start of a new treatment. All imaging was performed before start and during treatment with afatinib. We therefore did not have imaging after cessation of therapy (as patients were operated shortly after cessation of TKIs). As we did not have access to surgery or histopathology reports we cannot confirm (nor exclude) if any of the patients showed hyper progression shortly after discontinuation of afatinib.

Finally, again, in the discussion you should emphasize the clinical implications, in a "practical" way of adding the results of a radiomics analysis to the management of these patients.

Thank you for the suggestion. The section on the semantics of the selected radiomic features was extended. More information was provided on why sphericity of tumour diameter may be important prognostic factors. Also, we elaborated on the use of specific cut-offs for (a)sphericity and diameter in clinical practice. Please refer to line 269-283:

“A lower sphericity, i.e., a larger surface compared to the volume, is associated with higher mortality. It is hypothesised that the sphericity of the tumour impacts its resectability: a perfect spheric tumour might be easier to resect than aspheric tumour with a large surface area compared to the volume. Several previous studies already described the prognostic value of primary tumour asphericity in (oral cavity) HNSCC determined on [18F]FDG PET [33,34] and CT [35], including cut-offs for the prediction of overall and progression-free survival. Please note that the definition of sphericity and asphericity may vary between software packages and cut-offs might not be applicable for each software implementation. Preferably, feature definitions of the Image Biomarker Standardisation Initiative are used [36].

Also, both models selected a 2D diameter feature from the evaluation PET, which is defined as the largest pairwise Euclidian distance in a specific direction. The 2D diam-eter slice represents the diameter in the axial plane and the 2D diameter column rep-resents the coronal plane. This indicated that a larger tumour diameter, specifically in the transversal plane, is associated with higher mortality. Tumour size is considered an important prognostic factor in HNSCC [22] [37]. Our findings are similar to those of Apostolova at al., as they present an incremental prognostic value of metabolic tumour volume (tumour dimension in 3D) in addition to asphericity [33].”

Reviewer 2 Report

The study demonstrates a radiomic model for the overall survival of head and neck cancer patients receiving afatinib based on data from two centres. I have the following comments.

The proportional hazard assumption has to be checked before applying Cox regression.

It is suggested to use a machine learning algorithm to predict the survival time, which is expected to associate with better performance.

It is suggested to extract quantitative radiomic features to explore the possibility of better model performance.

  • The poor prognosis of head and neck cancer, which calls for better diagnosis and treatment plan, is elaborated (PMID: 29447088, PMID: 31766180). Please discuss.

Author Response

Reviewer 2:

The study demonstrates a radiomic model for the overall survival of head and neck cancer patients receiving afatinib based on data from two centres. I have the following comments.

Thank you for your comments regarding our article. We appreciate your time and effort reviewing our article and your suggestions. Thanks to your comments, our manuscript has greatly improved.

The proportional hazard assumption has to be checked before applying Cox regression.

Thank you for pointing this out. We assessed the proportional hazard assumption graphically by plotting the Schoenfeld residuals against time. We have added these plots in supplementary figure 1. The plots show that all the covariates in the Cox proportional hazards model met the proportional hazard assumption.

We have added:

“The proportional hazard assumption was graphically assessed by plotting the Schoenfeld residuals against time.” Line 195-196.

“Plotting the Schoenfeld residuals against time showed that all the covariates in the Cox proportional hazards model met the proportional hazard assumption (supple-mentary figure 1).” Line 207-209.

“Supplementary figure 1. Proportional hazard assumption test for overall survival by plotting the Schoenfeld residuals against time for both the EORTC and Unicancer cohort. The x-axis represents the survival time in months; the y-axis represents the scaled Schoenfield residuals for sphericity, maximum 2D diameter slice/column, age, and N-stage.”

It is suggested to use a machine learning algorithm to predict the survival time, which is expected to associate with better performance.

Thank you for the suggestion. We have considered other machine learning algorithms for modelling like LASSO or elastic net regularised generalised linear models. However, we chose to use a Cox proportional hazards model because it is a (clinically) well-established method for analyzing time-to-event data. The Cox model allows us to estimate the effect of various covariates on survival outcomes while accounting for the time-to-event nature of the data. In addition, LASSO regularization might be useful for variable selection and reducing overfitting in linear models, but since we already performed strict feature selection, we did not opt for this. Also, a recent publication by Amini et al. (PMID: 34872823) shows no significant differences in model performances between a Cox model and regularized linear model for PET radiomics. Therefore, the model with best (clinical) explainability was chosen.

It is suggested to extract quantitative radiomic features to explore the possibility of better model performance.

We would like to request further clarification as radiomic features are inherently quantitative by nature. Radiomic features were extracted from [18F]FDG PET, an imaging modality that as no other, allows for quantification of [18F]FDG uptake.

The poor prognosis of head and neck cancer, which calls for better diagnosis and treatment plan, is elaborated (PMID: 29447088, PMID: 31766180). Please discuss.

Thank you for the great suggestion to add more information on the poor prognosis of HNSCC. At the request of reviewer 1, more information on the prognosis and main critical issues of HNSCC were added to line 68-78 of the introduction. Also, we added more information to line 253-257 of the discussion including (PMID: 29447088, PMID: 31766180).

“HNSCC is a challenging disease to treat, particularly when it is diagnosed at an advanced stage, and early diagnosis and intervention are crucial for improving patient outcomes [3,22]. Radiomics has the potential to play an important role in the manage-ment of HNSCC by identifying patients who may benefit from a specific therapy, evaluate the risk of recurrence, and to predict survival.

Round 2

Reviewer 1 Report

The manuscript has been greatly improved now and can be considered for publication.